# Universal Properties of Activation Sparsity in Modern Large Language Models

**Filip Szatkowski[1,2]**      **Patryk Będkowski[1]**      **Jan Dubiński[1,3]**

**Alessio Devoto[4]**      **Pasquale Minervini[5,6]**      **Mikołaj Piórczyński[1,7]**

**Simone Scardapane[4]**      **Bartosz Wójcik[8,9]**

## Abstract

Input-dependent activation sparsity is a notable property of deep learning models, which has been extensively studied in networks with ReLU activations and is associated with efficiency, robustness, and interpretability. However, the approaches developed for ReLU-based models depend on exact zero activations and do not transfer directly to modern large language models (LLMs), which have abandoned ReLU in favor of other activation functions. As a result, current work on activation sparsity in LLMs is fragmented, model-specific, and lacks consensus on which components to target. We propose a general framework to assess sparsity robustness and present a systematic study of the phenomenon in the FFN layers of modern LLMs, including diffusion LLMs. Our findings reveal universal patterns of activation sparsity in LLMs, provide insights into this phenomenon, and offer practical guidelines for exploiting it in model design and acceleration.

## 1 Introduction

An intriguing property of deep learning models is activation sparsity [22], i.e., the tendency of hidden states to contain mostly zero (or near-zero) values with input-dependent patterns. This phenomenon can be used to improve model efficiency through skipping computations [25, 28], and has been linked to robustness [7, 22] and interpretability [2, 4, 11, 14, 48]. Activation sparsity has been extensively studied for ReLU-based networks, including MLPs [1, 15], CNNs [17, 33], and vanilla Transformer feed-forward networks (FFN) [22, 47]. Modern LLMs [5, 6, 13, 27, 31, 32, 42], however, use SiLU/GELU activations and GLU-based FFNs [34], and do not produce exact zero activations.

Although activations of such LLMs still exhibit substantial sparsity, approaches developed to study and exploit it in ReLU-based networks often fail to transfer directly [10, 38, 41]. While some works explored retrofitting LLMs to use ReLU activations to introduce exact sparsity [28, 37, 38, 47], such approaches sacrifice model quality [10, 37] and come with additional costs and constraints.

Therefore, recent works developed tailored techniques to exploit activation sparsity for acceleration of modern LLMs [3, 10, 18, 23, 25]. However, these methods are typically model- or module-specific and require extra steps such as additional training [28, 41], sparsity prediction [25, 47], or calibration on held-out data [18, 23]. There is also no consensus on which FFN components to focus on, with approaches targeting inputs [10, 23, 24], gate [18], or intermediate states [25]. Given these rapid advances, we argue that a systematic study of activation sparsity in modern LLMs can

---

[1]Warsaw University of Technology, [2]IDEAS NCBR, [3]NASK National Research Institute, [4]Sapienza University of Rome, [5]University of Edinburgh, [6]Miniml.AI, [7]IDEAS Research Institute, [8]Jagiellonian University, [9]Tooploox

Preprint.

provide valuable insights into their inner workings, network embedding geometry, and model design. Therefore, we provide a consolidated discussion of activation sparsity characteristics in widely used LLMs and propose a simple and universal approach to determine the robustness of Transformer activations to sparsity. Using this framework, we analyze sparsity patterns across FFN components, model families, and sizes, and examine how data and training affect sparsity robustness. Finally, we discuss how our findings relate to the existing body of work on activation sparsity. Our work highlights universal patterns in activation sparsity and provides practical guidelines for practitioners seeking to understand activation sparsity or leverage this curious phenomenon for model acceleration.

## 2 Activation Sparsity in Modern Transformers

**Gated Feedforward Transformer Layers.** Transformer blocks consist of attention and feed-forward (FFN) sub-blocks [43]. While the FFN in the original Transformer was composed of two projection layers separated by an activation function, modern LLMs typically employ FFNs based on the Gated Linear Unit (GLU) architecture [34], which can be expressed as: $\mathcal{FFN}(x) = \mathbf{W_d}\big((\mathbf{W_u}x) \odot \sigma(\mathbf{W_g}x)\big)$, where: $x \in \mathbb{R}^h$ is the input vector, $\mathbf{W_u} \in \mathbb{R}^{h \times d}$ is the up-projection matrix, $\mathbf{W_g} \in \mathbb{R}^{h \times d}$ is the gating projection matrix, $\mathbf{W_d} \in \mathbb{R}^{d \times h}$ is the down-projection matrix, and $\sigma$ is an activation function, usually SiLU or GELU. We use $h$ and $d$ to denote the model's hidden and intermediate dimensions, respectively. In the subsequent sections, we refer to the above-mentioned activation vectors in the FFN as $x$ - *input* , $u = \mathbf{W_u}x$ - *up-projection* ,

$g = \sigma(\mathbf{W_g}x)$ - *gate* and $i = (\mathbf{W_u}x) \odot \sigma(\mathbf{W_g}x)$ - *intermediate* vectors.

**Computational Benefits of Activation Sparsity.** Activation sparsity refers to models relying on a small, input-dependent subset of neurons and leaving most activations effectively unused. This enables skipping parts of FFN matrix multiplications, reducing both computation and weight loading costs (Figure 1), with potential additional hardware-specific gains [10]. Activation sparsity approaches usually differ along two axes: 1) value-based vs. predictor-based: directly using activations to skip computation in other modules versus predicting masks with auxiliary networks, and 2) column-wise vs. row-wise: skipping computation in the matrix columns or in the matrix rows. Value-based approaches usually either sparsify the computation column-wise in the linear layers based on their input [10, 23, 24] or use gate values to determine which computation to skip in a row-wise manner in up- and down-projection layers [18]. Predictor-based approaches focus on determining sparsity in the intermediate activations [25, 41, 47], which allows for row-wise sparsification of all three FFN matrices. While these methods potentially achieve much higher sparsity, these gains come at the cost of additional predictor computations.

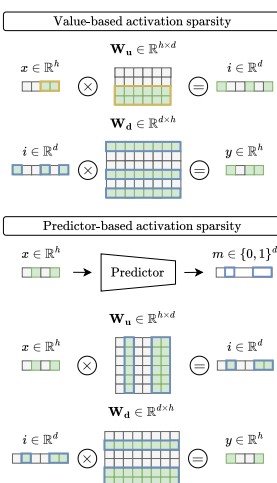

Figure 1: Model acceleration approaches.

**Determining Activations to Sparsify in Non-ReLU LLMs.** Modern LLM architectures lack components that explicitly produce zero activations, which makes it difficult to study activation sparsity directly. However, prior work [3, 23, 24] has shown that some activation vectors $v \in \mathbb{R}^n$ in such models can be *sparsified* to a certain degree without incurring a significant performance loss. To study the general impact of sparsification on FFN layers, we propose to use a simple top-p sparsification rule, where we obtain a sparsity mask $m_p$ from the largest-magnitude entries in $v$ whose absolute values sum to at least a fraction $p$ of the vector's total L1 norm:

$$\text{top-p}(v) = m_p \odot v; \ m_p = \underset{m}{\arg\min} ||m||_0 \ \text{s.t.} \ ||m \odot v||_1 \geq p \cdot ||v||_1 \ \text{and} \ m \in \{0, 1\}^n.$$

The induced sparsity is then the fraction of zeros in $m_p$. By evaluating model performance over a range of $p$ values, we can obtain a sparsity-performance trade-off curve and assess the functional activation sparsity of the model – the level of sparsity at which the model still performs similarly to the densely activated original. Our approach is simple, general, easy to interpret, and provides a reasonably good activation sparsity in practice. Crucially, it can be applied to any FFN module without auxiliary training or calibration, which allows us to fairly compare models and modules. See Appendix B for further discussion and comparison between the top-p approach and the alternatives.

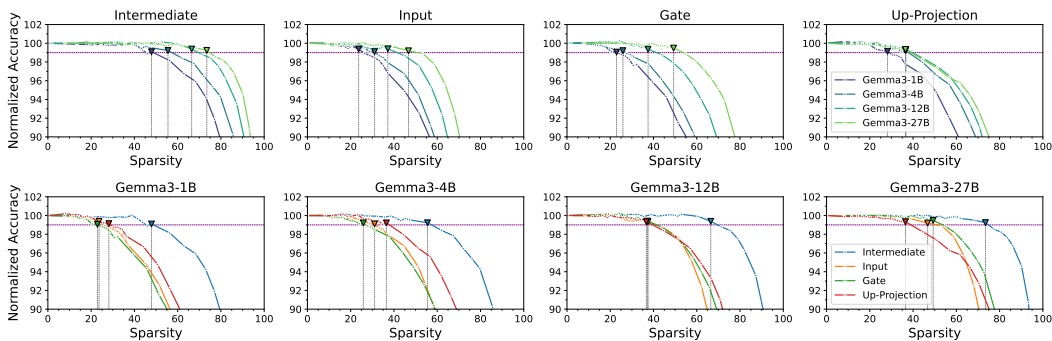

Figure 2: Average accuracy across downstream tasks normalized by the original performance with different induced activation sparsity for base Gemma3 models. (top) Sparsity for different FFN modules at various model sizes. (bottom) Sparsity for different models at various modules. We denote the highest (*critical*) sparsity where at least 99% performance is retained with a marker.

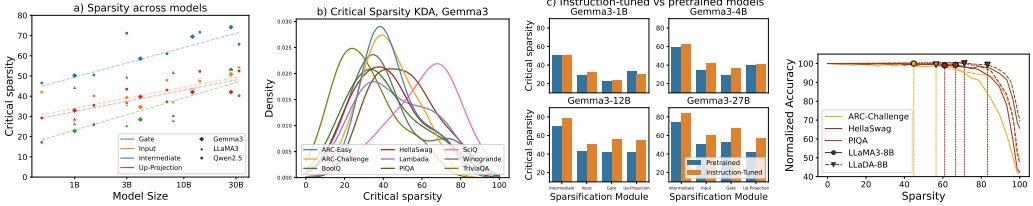

Figure 3: a) Critical sparsity across model families, scales, and modules. b) Per-task sparsity KDA across Gemma3 experiments. c) Critical sparsities for pretrained and Instruction-Tuned Gemma3 models. d) Activation sparsity with LLaMA-8B and LLaDA-8B, with critical sparsity points marked.

## 3  Experiments

To study the effects of activation sparsity in LLMs, we use the task suite from Mirzadeh et al. [28] and evaluate Gemma3, Llama3.1/3.2, and Qwen2.5 models with `lm-eval-harness` [12] in a zero-shot setting. For each experiment, we fix a threshold $p$ and apply the top-$p$ rule uniformly to one of four activation types across all model layers. We then measure the average induced sparsity and the resulting performance drop. Unless stated otherwise, we report the average sparsity and performance across all the tasks. To explicitly tie sparsity with accuracy, we use the concept of *critical sparsity* - the highest empirical sparsity level at which models still retain at least 99% of their accuracy.

**Which parts of the FFN are most robust to sparsity?**   We evaluate Gemma3 models by plotting performance degradation under increasing sparsity in Figure 2. **Sparsity robustness generally improves with model size**, barring small fluctuations in up-projections. Although intermediate activations show the greatest sparsity, they offer limited use for FFN acceleration, since they directly allow sparsifying around one-third of FFN operations. However, **intermediate activations allow for the highest efficiency gains when sufficient predictors are available**. Despite its simplicity, the input-based method achieves sparsity levels comparable to the gate-based approach often favored in the literature [18]. Gate sparsity is typically no higher than input sparsity, and up-projection sparsity performs similarly to gates, despite the latter applying an activation function. **Input-based sparsity appears the most practical for predictor-free methods**, as it matches gate sparsity while allowing the acceleration of all FFN modules. Gate-based sparsification, contrary to intuition, offers no clear advantage at our scale, though for models larger than ∼30B parameters it may surpass input sparsity.

**How does activation sparsity behave across models?**   To assess the generality of our previous findings across families and scales, we consider pretrained LLaMA3.1/3.2 and Qwen2.5 models and plot their critical sparsity in Figure 3a, fitting trends with least squares (see Appendix A for numerical results). **The trends outlined in the previous section remain roughly consistent across models**: intermediate activations are generally the most sparse, with input and up-projection activations achieving higher sparsity than the gate until the larger model sizes. Slight deviations in the trends can be attributed to non-uniform depth–width scaling, especially in Qwen, where dimensions grow

disproportionately with parameter count. Overall, **activation sparsity tends to increase with model size**, though it cannot be directly determined based on the model size alone.

**Is activation sparsity task-dependent?**  Activation sparsity patterns are dynamic and input dependent, raising the question of whether robustness also varies across tasks. To examine this, we analyze the kernel density estimate (KDA) [36] of critical sparsities obtained for different tasks across Gemma3 modules and model sizes in Figure 3b. Critical sparsity differs widely across the tasks, which indicates that **the phenomenon of activation sparsity is also highly task dependent**. This supports prior work advocating task-specific acceleration approaches [8, 49].

**Does training affect activation sparsity?**  Critical sparsity depends on the model architecture and size, sparsification method, and task. An interesting question is also whether changing the model training recipe affects the sparsity. To investigate this, we compare the average performance of pretrained and instruction-tuned Gemma 3 models in Figure 3c. At larger sizes, instruction-tuned models show higher tolerance to activation sparsity, indicating that **training influences robustness to sparsity** even with identical architectures. We observe variance between instruction-tuned and pretrained models across all the evaluated architectures (see Appendix A for numerical results), which aligns with prior work suggesting that training schemes significantly influence model robustness [16, 39] and underscores activation sparsity as a complex, training-dependent phenomenon.

**Is activation sparsity also prevalent in diffusion LLMs?**  Investigating training dependence further, we ask whether activation sparsity also arises in diffusion LLMs. While prior work has examined sparsity and caching in image diffusion [19, 26, 35, 46], to our knowledge, this is the first analysis of the phenomenon in diffusion-based LLMs (as discussed in Appendix C). We compare two models with identical architectures that differ by training paradigms: the masked diffusion LLaDA-8B [29] and the autoregressive LLaMA3.1-8B. Using our evaluation framework and the official LLaDA implementation, we apply independent sparsification at each diffusion step and evaluate on four different tasks. As shown in Figure 3d, LLaDA also exhibits significant activation sparsity, with even slightly more favorable sparsity–performance characteristics. Our findings suggest that **activation sparsity can also be a promising tool for accelerating diffusion LLMs**.

## 4   Discussion

Despite lacking any architectural bias toward explicitly sparse activations, modern LLMs consistently exhibit functional sparsity. **We argue that functional sparsity is a universal property of LLMs and advocate for its wider adaptation when designing efficient models.**

We find that larger models tend to exhibit higher sparsity, suggesting that frontier models will become sparser as scaling continues. Therefore, **activation sparsity stands out as a promising tool for accelerating ever-growing LLMs**, and we already see its adoption in models such as Gemma3n [45].

Our work is the first to examine functional sparsity in diffusion LLMs, a rapidly growing research area. We highlight sparsity as a promising avenue for improving their efficiency, and expect that **activation sparsity could see increasing adoption in diffusion LLMs as their development advances**.

Our results show that input activations match or exceed the sparsity of gates and up-projections. Computing gates to choose sparsity patterns [18] is wasteful if they are no sparser than inputs, and newer work [10, 23, 24] demonstrates stronger acceleration with purely input sparsity. Overall, **our results suggest that input sparsification is the most efficient approach**.

The high variance of critical sparsity across evaluation tasks and training recipes calls into question methods that rely on extra training [25, 41, 47] or threshold calibration [18, 23, 24] on auxiliary datasets. Our results suggest that **sparsification methods should be truly data-free, as both functional sparsity levels and resulting patterns can be prone to overfitting**.

Our results should be seen as a lower bound on activation sparsity, as we adopt a simple, broadly applicable framework. While layer- or module-specific methods may achieve higher sparsity, our top-$p$ approach already reaches practical levels comparable to existing work. Given this and our earlier arguments on overfitting, **we argue that sparsification method design should favor simplicity**.

We follow the evaluation setting of Mirzadeh et al. [28] and focus on likelihood-based evaluations of pretrained LLMs. Although we do not test reasoning models directly, **the consistency of our**

**findings across instruction-tuned and diffusion LLMs strongly suggests that activation sparsity will also benefit reasoning models**, which rely on the same architectures as studied in our work.

Finally, we emphasize that **activation sparsity should be viewed as complementary to other acceleration methods** such as quantization or speculative decoding. FFN sparsity can only be pushed to moderate levels before performance degrades, capping efficiency gains at about 1.3–1.5x [18, 23, 24], far below 4x speedups achievable with other methods [20, 21]. Given this, **we argue that evaluations of activation sparsity methods should prioritize performance preservation**, since degradation occurs at very different levels depending on the model and task. We therefore advocate for a focus on reachable sparsity that does not harm performance, as shown in our notion of critical sparsity.

We hope our work sheds light on the universal phenomenon of activation sparsity in LLMs, characterizes its potential for practical acceleration, and provides useful insights for future model design.

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

# Appendix

## A    Critical Activation Sparsity of Pretrained and Instruction-Tuned models

In Table 1, we report the exact numerical values of the critical activation sparsity for all models considered in our experiments, including both pretrained and instruction-tuned variants. The metrics $S_{inter}$, $S_{input}$, $S_{gate}$, and $S_{up\_p}$ denote the critical sparsity levels for the intermediate, input, gate, and up-projection activations, respectively.

For completeness, we also include key model hyperparameters such as the number of layers ($N_L$), hidden dimension ($dim_h$), and intermediate dimension ($dim_i$). While we do not observe clear, direct relationships between these hyperparameters and the achieved critical sparsity, the general trend of sparsity increasing with model size remains evident. Notably, the Qwen family exhibits some fluctuations, which may stem from the non-uniform scaling of its architectural hyperparameters across model sizes.

Table 1: Critical activation sparsity for pretrained and instruction-tuned models. $S_{inter}$, $S_{input}$, $S_{gate}$, and $S_{up\_p}$ refer to intermediate, input, gate, and up-projection activation sparsity, respectively.

| Model | $N_L$ | $dim_h$ | $dim_i$ | Pretrained | | | | Instruction-Tuned | | | |
|---|---|---|---|---|---|---|---|---|---|---|---|
| | | | | $S_{inter}$ | $S_{input}$ | $S_{gate}$ | $S_{up\_p}$ | $S_{inter}$ | $S_{input}$ | $S_{gate}$ | $S_{up\_p}$ |
| Gemma3-1B | 26 | 1152 | 6912 | 50.22 | 28.53 | 22.83 | 32.96 | 49.98 | 32.14 | 23.23 | 30.27 |
| Gemma3-4B | 34 | 2560 | 10240 | 58.56 | 34.63 | 28.50 | 39.72 | 62.82 | 42.29 | 35.99 | 40.82 |
| Gemma3-12B | 48 | 3840 | 15360 | 69.46 | 43.03 | 42.05 | 42.03 | 78.77 | 50.74 | 55.45 | 54.26 |
| Gemma3-27B | 62 | 5376 | 21504 | 74.12 | 50.83 | 53.03 | 42.01 | 84.05 | 59.88 | 68.15 | 56.95 |
| LLaMA3.2-1B | 16 | 2048 | 8192 | 44.44 | 28.09 | 26.51 | 28.82 | 45.02 | 29.65 | 24.30 | 29.70 |
| LLaMA3.2-3B | 28 | 3072 | 8192 | 49.58 | 35.91 | 25.52 | 37.14 | 58.07 | 33.28 | 28.90 | 44.72 |
| LLaMA3.1-8B | 32 | 4096 | 14336 | 51.89 | 37.31 | 28.04 | 30.52 | 61.96 | 39.34 | 34.38 | 41.76 |
| Qwen2.5-0.5B | 24 | 896 | 4864 | 46.54 | 42.01 | 17.16 | 29.20 | 43.92 | 32.80 | 24.60 | 32.12 |
| Qwen2.5-1.5B | 28 | 1536 | 8960 | 50.49 | 40.12 | 25.93 | 35.50 | 52.93 | 32.50 | 27.63 | 32.99 |
| Qwen2.5-3B | 36 | 2048 | 11008 | 71.16 | 39.40 | 39.58 | 43.46 | 59.80 | 44.16 | 36.90 | 36.32 |
| Qwen2.5-7B | 28 | 3584 | 18944 | 60.98 | 47.89 | 37.25 | 43.05 | 59.95 | 47.01 | 32.58 | 40.62 |
| Qwen2.5-14B | 48 | 5120 | 13824 | 71.66 | 47.39 | 48.04 | 52.25 | 69.35 | 50.10 | 41.87 | 49.04 |
| Qwen2.5-32B | 64 | 5120 | 27648 | 65.66 | 54.08 | 40.20 | 52.46 | 68.77 | 55.17 | 40.54 | 57.35 |

# B How To Induce Activation Sparsity in FFNs?

## B.1 Alternative sparsification rules

In Section 2, we propose to use the top-p sparsification rule to induce the sparsity in the activation vectors of the models. We opt for a simple sparsification rule to avoid any data dependency or bias towards a specific model or FFN module. However, top-p is not the only possible way to perform sparsification, and many other works opted for alternative methods to extract the sparse subsets of neurons, such as top-k [47] or max-p [41].

Assuming vector $v \in \mathbb{R}^n$, top-k finds $k$ largest neurons in the vector and can be formally defined as a transformation that multiplies $v$ with a subset of $k$ neurons which maximizes the norm of the sparsified vector:

$$\text{top-k}(v) = m_k \odot v; \; m_k = \underset{m}{\text{argmax}} \, ||m \odot v||_1 \; \text{ s.t. } ||m||_0 = k \text{ and } m \in \{0,1\}^n.$$

Similarly, max-p finds the subset of the neurons that satisfy the condition that their absolute values are at least $p \cdot \max(v)$:

$$\text{max-p}(v) = m_p \odot v; \; m_p = \underset{m}{\text{argmin}} \, ||m||_0 \; \text{ s.t. } |v_i| \cdot m_i \geq p \cdot ||v||_\infty \; \forall i \text{ and } m \in \{0,1\}^n,$$

where $||v||_\infty = \max_i |v_i|$ denotes the maximum absolute entry of $v$, and the mask $m_p$ retains exactly those coordinates $i$ for which $|v_i| \geq p \cdot ||v||_\infty$. Notably, the mask always selects the largest entry in the activation vector.

We empirically compare the three sparsification strategies in Figure 4, focusing on the sparsity–accuracy tradeoff averaged over our evaluation tasks for the smallest model in each family. Overall, top-p and top-k produce very similar curves, whereas max-p underperforms in certain settings. Therefore, we adopt top-p for our experiments, as it is more interpretable than top-k and can more universally transfer across model sizes. In particular, larger models typically yield higher sparsity, requiring $k$ to be carefully chosen as most values of $k$ have no effect until a critical sparsity is reached. By contrast, with top-p performance degrades more smoothly and predictably, allowing us to evaluate a fixed set of thresholds that transfer well across models.

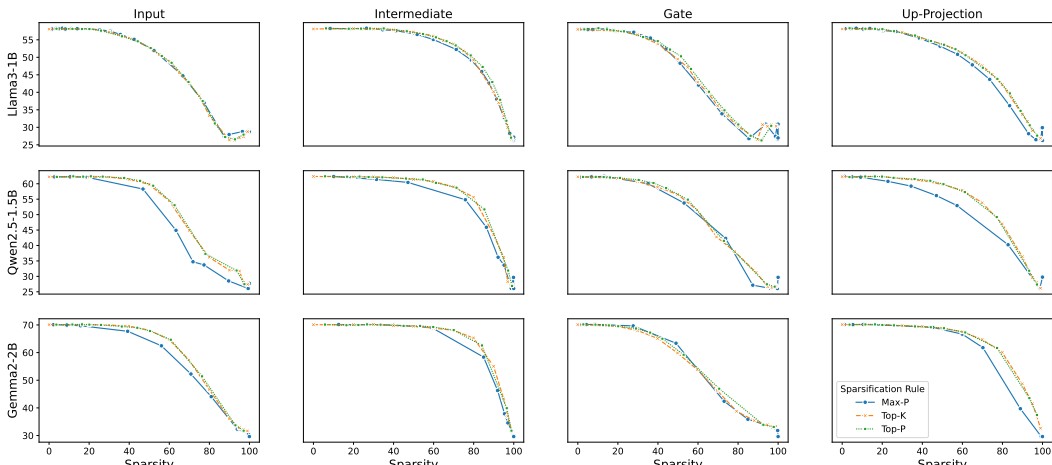

Figure 4: Comparison of sparsification rules for different models and different blocks.

## B.2 Sparsification rule transferability between the models

To further study the transferability and impact of the threshold selection in different models, we investigate the activation sparsity induced in separate layers of Gemma3 and Qwen2.5 models. We select a subset of $p$ thresholds, and register the activation sparsity obtained at a given layer alongside the average of the accuracy under the threshold. We plot the results as heatmaps in Figures 5 and 6.

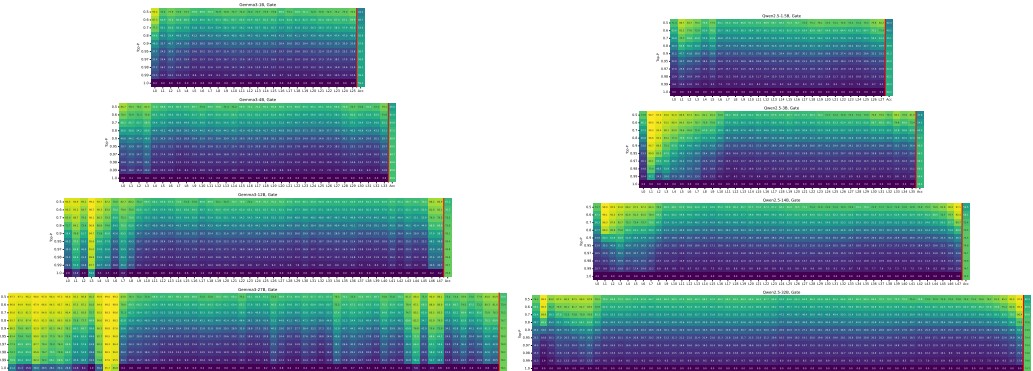

Figure 5: Top-p threshold values and resulting sparsity induced in the gate activation vectors alongside the accuracy with the given threshold across different layers of the Gemma3 and Qwen2.5 models.

First, we investigate the sparsity of gate activations in the Gemma and Qwen models of corresponding sizes in Figure 5. Except for some early layers, the sparsity values obtained across the models appear similar for a given threshold across the middle layers.

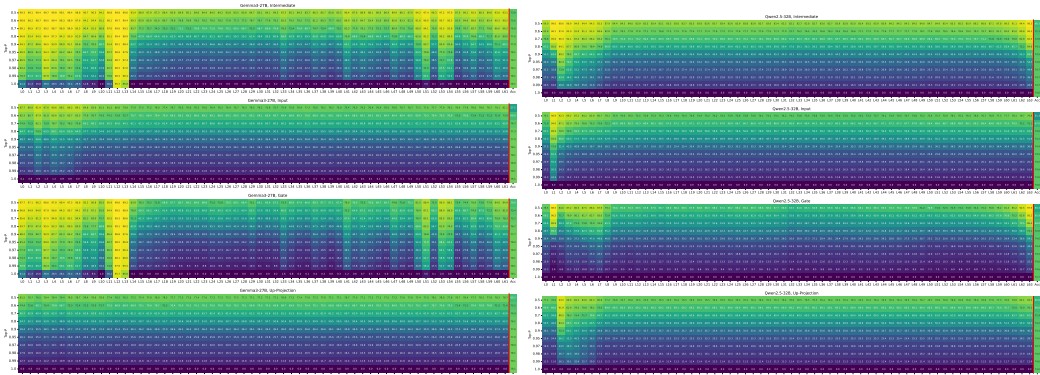

Figure 6: Top-p threshold values and resulting sparsity for Gemma3-27B and Qwen2.5-32B models.

Then, in Figure 5 we plot the sparsity of all four investigated activation vector types in the largest models within each model family. Again, except for a few layers, the sparsities obtained for a given $p$ appear similar between the models.

Both of these results support the universality of our approach and our decision to choose $\mathrm{top\text{-}p}$ over $\mathrm{top\text{-}k}$, as using $\mathrm{top\text{-}k}$ would require more manual threshold selection to find the critical sparsity, as outlined by the variance in the critical sparsity across different model sizes and types in Section 3. While the resulting sparsity heatmaps show a few high outliers, particularly for Gemma3-27B gates, we attribute the presence of these to the presence of massive activations [40], as for massive outlier values, the magnitude of the vector norms that we use will concentrate around very few large values and may even cause exact sparsity to appear at $p = 1.0$ as the nature of the numerical precision will make the smallest entries in the activation vector appear like zeros since they basically contribute nothing compared to the massive outlier. We do not investigate this phenomenon further and leave it for future work. However, we note that it can have important implications for the design of activation sparsity approaches, particularly those that rely on thresholding, as rules and thresholds devised for such massive activations might be highly unstable when encountering out-of-distribution data.

# C  Activation Sparsity for the Acceleration of Diffusion Models

Recent work has revealed strong activation sparsity and temporal redundancy in diffusion models across modalities. In text-to-image models like Stable Diffusion, DeepCache [26] leverages the fact that many neuron activations and feature maps change little between denoising steps by reusing high-level U-Net activations across timesteps, achieving over 2x speedups with minimal quality loss. Similarly, Chipmunk [35] applies the same idea to diffusion Transformers, caching activations and updating only the small set of neurons that change, which enables up to 3.7x faster video generation. However, Zhang et al. [46] highlight the risks of naïve caching, showing that it can degrade diversity. Their Dynamics-Aware Token Pruning (DaTo) method selectively updates only tokens with meaningful changes, preserving quality while achieving 7–9x acceleration. Together, these results suggest that only a small, stable subset of neurons or tokens drives most of the generative process in diffusion models for vision.

Sparsity has also been observed in the weights of diffusion models. Fang et al. [9] identify redundant parameters over time using a Taylor-based method, pruning up to 50% of weights with minimal quality degradation. Structured sparsity approaches such as SparseDM [44] and sparse-to-sparse training [30] further demonstrate that diffusion models with 50–80% weight sparsity can match or even outperform dense models, indicating the presence of robust sparse subnetworks.

While, to our best knowledge, neuron-level sparsity in diffusion LLMs remains underexplored, early work by Li et al. [19] shows that temporal step pruning can significantly reduce the number of inference steps with little quality loss, achieving up to 400× speedups while preserving fluency. This points to significant redundancy in text diffusion models, similar to that observed in vision and video diffusion models, and further supports the adoption of activation sparsity for their acceleration. Taken together, prior work exploiting activation sparsity for acceleration, pruning, and compression and our own analysis in Section 3 suggest that activation sparsity is a promising future direction for acceleration of diffusion LLMs.

