# OpenReview forum: "Universal Properties of Activation Sparsity in Modern Large Language Models"
_NeurIPS.cc/2025/Workshop/UniReps — UniReps2025_

### Official Review · Reviewer_Xjmf · 2025-09-06
**Universal Properties of Activation Sparsity in Modern Large Language Models**

**Confidence:** 4

**Review:**

## Paper Summary
The paper investigates activation sparsity in modern non-ReLU LLMs using a proposed simple *top-p sparsification* framework. The authors evaluate sparsity robustness across Gemma, LLaMA, Qwen, and diffusion LLMs, arguing that activation sparsity is a *universal property* with implications for efficiency and model design.

## Quality
The experiments are thorough and span several model families; however, the methodology is simple and largely descriptive. The top-p rule provides a baseline characterization but is not a fundamentally new approach.

## Clarity
The paper is well-written, with a clear structure and informative figures. However, a deeper analysis would be welcomed.

## Originality
The novelty is limited: top-p sparsification is straightforward, and much of the contribution is empirical consolidation of earlier studies. The extension to diffusion LLMs is a notable addition, but it is too qualitative.

## Significance
The practical impact appears limited. Although reported efficiency gains are not bad,  the study does not offer a clear path toward surpassing other acceleration techniques, such as quantization or speculative decoding. This experiment addition would strengthen the contributions. Furthermore, the universality claim lacks a deeper theoretical justification, leaving open the question of why functional sparsity emerges.

## Pros
* Systematic empirical study across several model families.
* Clear presentation.
* Extension to diffusion LLMs is an interesting contribution.

## Cons
* Methodological novelty is minimal (the top-p rule is simple and unsurprising).
* No concrete acceleration strategies proposed.
* The universality claim is asserted but not theoretically grounded.

## Overall
This work provides a careful and well-structured empirical investigation. Its main value lies in consolidating fragmented observations into a single, coherent study. However, the methodological novelty is limited, efficiency gains are modest, and the universality claim would benefit from stronger theoretical grounding. Therefore, I see this paper as borderline: potentially useful to the community as a reference, but not a particularly strong contribution in terms of innovation.

**Score:**

3

**Topic Fit:**

3

---

### Official Review · Reviewer_9p87 · 2025-09-10
**Review of Universal Properties of Activation Sparsity in Modern Large Language Models**

**Confidence:** 3

**Review:**

This paper presents a systematic study of activation sparsity in modern Large Language Models (LLMs) that use SiLU/GELU activations and GLU-based FFNs, rather than ReLU.

Strengths
- Simple and Universal Framework: The proposed top-p sparsification method requires no additional training or calibration, and enables fair comparisons across models and modules. The "critical sparsity" metric (99% performance retention) provides a clear and interpretable measure.
- Comprehensive Experimental Coverage: The experiments span multiple dimensions - different FFN components (input, gate, up-projection, intermediate), model sizes, training recipes (pretrained vs instruction-tuned), and even extends to diffusion LLMs (LLaDA), which is novel.
- Clear Presentation: The paper is well-written with effective visualizations (particularly Figures 2 and 3) that clearly communicate the main findings.

Weaknesses
- Narrow Evaluation Scope: The evaluation focuses solely on likelihood-based tasks from Mirzadeh et al. [27]. Missing are: (a) generation tasks with actual text generation quality metrics, (b) reasoning benchmarks (GSM8K, MATH), (c) long-context evaluations where sparsity patterns might differ significantly.
- Insufficient Comparison with Existing Methods: While the paper criticizes predictor-based and calibration-based approaches, it doesn't provide direct comparisons with state-of-the-art methods, or recent work on dynamic sparsity. The claimed "lower bound" on activation sparsity needs empirical validation.
- The diffusion LLM section feels rushed. How does sparsification at each diffusion step affect the denoising process? Are early vs. late diffusion steps equally robust to sparsity?
- What happens with different p-norm choices? How sensitive are results to the 99% threshold? How do results change with different task weights in the average?

Minor Issues
Line 75: The mathematical notation could be cleaner with proper set notation
Figure 1 caption is too brief and doesn't explain the visualization adequately
The paper uses "critical sparsity" inconsistently - sometimes as a concept, sometimes as a specific metric

**Score:**

3

**Topic Fit:**

2

---

### Official Review · Reviewer_SET3 · 2025-09-14
**Concise, constructive review of “Universal Properties of Activation Sparsity in Modern Large Language Models”**

**Confidence:** 4

**Review:**

The paper studies input-dependent activation sparsity in modern LLM FFN layers (GLU/SiLU/GELU; non-ReLU). It introduces a calibration-free top-p sparsification probe to trace accuracy–sparsity trade-offs across activation sites (input, gate, up-projection, intermediate), model families/sizes (Gemma, Llama, Qwen), and training regimes (pretrained vs. instruction-tuned), plus an initial look at a diffusion LLM (LLaDA). A key notion is critical sparsity: the highest sparsity with ≥99% retained accuracy. Findings: (i) functional sparsity is widespread and generally increases with scale; (ii) input activations are as sparse as (or sparser than) gates, making them attractive for predictor-free speedups; (iii) sparsity is task-dependent and training-dependent; (iv) diffusion LLMs also exhibit meaningful sparsity.

Strengths

Timely, relevant scope: Activation sparsity is a hot area for inference efficiency and interpretability. A cross-family, cross-scale view is valuable for UniReps.
Simple, general probe: The top-p L1 mask is architecture-agnostic, enabling apples-to-apples comparisons without retraining or calibration.
Clear operational metric: Critical sparsity (≥99% accuracy) offers a practical, conservative target for safe sparsification.
Actionable guidance: Evidence that input-based sparsification can match gate-based methods supports simpler pipelines and may avoid wasted gate computation
Breadth of evaluation: Multiple sizes (1B→27B/32B) and families, pretrained vs. instruction-tuned variants, plus a first pass on a diffusion LLM.
Well-curated related work bridging efficiency, interpretability, and sparsity prediction.

Weaknesses / Limitations

Missing system-level evidence: The work largely reports functional sparsity vs. actual wall-clock speedups, bandwidth/DRAM reads, or kernel-level utilization. Even small microbenchmarks (per-layer GEMM skip, mask application overheads) on one GPU/CPU would strengthen practicality claims.
Top-p design choices: While intuitive, top-p can bias toward heavy-tailed coordinates; the paper compares to top-k/max-p, but it would help to analyze mask stability across tokens/layers and the cost of computing masks at runtime.
Task coverage vs. generality: lm-eval zero-shot suite is standard but mostly short-context; sparsity behavior on long-context or reasoning chains could differ. No results on multilingual/code or safety-aligned models.
Diffusion LLM section is thin: Interesting early result, but limited scope (one pair of models, 4 tasks). It’s unclear how diffusion-step coupling affects mask consistency or whether per-step independence leaves speed on the table.
Statistical treatment: Figures are persuasive, but variance/CI reporting is sparse; significance of trends (e.g., input≥gate) could be made more rigorous.
No fairness/robustness stress tests: Since sparsity ties to robustness in prior work, adversarial or distribution-shift probes would add depth.

**Score:**

4

**Topic Fit:**

3

---

### Official Review · Reviewer_WaU3 · 2025-09-14

**Confidence:** 3

**Review:**

The paper introduces a simple, model-agnostic way to quantify “functional” activation sparsity in GLU-based LLM FFNs using a top-p (L1-mass) masking rule, and summarizes performance with “critical sparsity” (max sparsity retaining ≥99% of baseline accuracy). Evaluating inputs, gates, up-projection, and intermediate activations across Gemma3, LLaMA3.x, Qwen2.5, and a diffusion LLM, they find: intermediate activations are most sparse but don’t directly speed up all FFN ops; input-based sparsity matches or exceeds gate-based and is the most practical predictor-free choice; sparsity generally increases with model size, varies significantly by task, and instruction-tuned models tolerate higher sparsity.

### Strengths

- Clear, general evaluation framework. The top-p (L1-mass) rule applies uniformly to any FFN activation without extra training or calibration, enabling fair cross-module/family comparisons via a single “critical sparsity” metric.
- Actionable design guidance. Inputs are at least as sparse as gates while enabling acceleration across all FFN matrices, making input-based sparsification the most practical predictor-free choice at current scales.


### Weaknesses
- Limited scope of evaluation and metrics. Results focus on accuracy under induced sparsity on zero-shot lm-eval tasks, without end-to-end latency/throughput measurements or direct tests on reasoning-tuned models.
- High task/training dependence and potential instability. Critical sparsity varies widely across tasks and training recipes, and massive-activation outliers can make thresholding brittle out-of-distribution, complicating deployment without per-task calibration.

**Score:**

3

**Topic Fit:**

1